# On Calibration of Multilingual Question Answering LLMs

**Yahan Yang**[*]                                                                                     *yangy96@seas.upenn.edu*
*University of Pennsylvania*

**Soham Dan**[*†]                                                                                    *sohamdan@microsoft.com*
*Microsoft*

**Dan Roth**                                                                                          *danr@seas.upenn.edu*
*University of Pennsylvania*

**Insup Lee**                                                                                          *lee@seas.upenn.edu*
*University of Pennsylvania*

**Reviewed on OpenReview:** *https://openreview.net/forum?id=4klghu2PTj*

## Abstract

Multilingual pre-trained Large Language Models (LLMs) are incredibly effective at Question Answering (QA), a core task in Natural Language Understanding, achieving high accuracies on several multilingual benchmarks. However, little is known about how well their confidences are calibrated. In this paper, we comprehensively benchmark the calibration of several multilingual LLMs (MLLMs) on a variety of QA tasks. We perform extensive experiments, spanning encoder-only, encoder-decoder, and decoder-only QA models (size varying from 110M to 7B parameters) and diverse languages, including both high- and low-resource ones. We study different dimensions of calibration in in-distribution, out-of-distribution, and cross-lingual transfer settings, and investigate strategies to improve it, including post-hoc methods and regularized fine-tuning. For decoder-only LLMs such as LlaMa2, we additionally find that in-context learning improves confidence calibration on multilingual data. We also conduct several ablation experiments to study the effect of language distances, language corpus size, and model size on calibration, and how multilingual models compare with their monolingual counterparts for diverse tasks and languages. Our experiments suggest that the multilingual QA models are poorly calibrated for languages other than English and incorporating a small set of cheaply translated multilingual samples during fine-tuning/calibration effectively enhances the calibration performance.

## 1 Introduction

Pre-trained Large Language Models (LLMs) like BERT, RoBERTA, T5, BART (Devlin et al., 2019; Liu et al., 2019; Wolf et al., 2020; Raffel et al., 2020; Lewis et al., 2019a), and their multilingual counterparts like mT5, mBART, XLM (Xue et al., 2021; Liu et al., 2020; Conneau et al., 2019), have greatly advanced Natural Language Understanding. The multilingual LLMs (MLLMs) are able to transfer knowledge acquired from English to other languages, leading to impressive cross-lingual performances on various downstream tasks in the zero-shot and few-shot settings, i.e., for languages not seen during fine-tuning (Wu & Dredze, 2020; Xue et al., 2021). Question Answering (QA) is a common task for understanding how well machines understand human language, and more importantly, it comes in a variety of formats such as multiple choice, span selection, or free-form answer generation. Several NLP/multimodal tasks can be cast as QA, highlighting the universality of the QA format. Multilingual QA as a task is becoming increasingly important with the globally widespread deployment and democratization of AI systems(Loginova et al., 2021).

---

[*] The first two authors contributed equally to this paper.
[†] Work done prior to joining Microsoft.

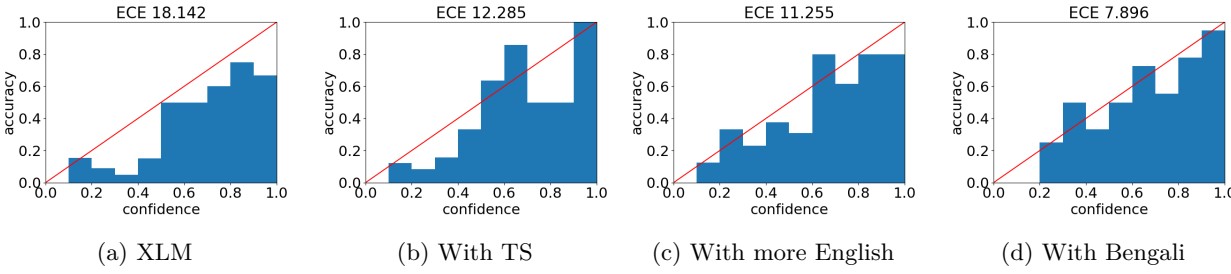

Figure 1: In this plot we show that pre-trained multilingual models, fine-tuned on English QA, are not well calibrated in languages other than English, specifically low-resource ones like Bengali. (a) shows the reliability diagram for XLM Conneau et al. (2019) fine-tuned on the English TyDiQA training set, evaluated on the Bengali TyDiQA test set. The large deviation from the diagonal Y=X line indicates it is not well calibrated. (b), (c) and (d) show that Temperature Scaling (TS), fine-tuning with more English data, and fine-tuning on Bengali TyDiQA training data respectively, all improve calibration compared to (a), as indicated by the better alignment with the diagonal and the lower ECE score. This indicates that despite high zero-shot cross-lingual accuracy, zero-shot cross-lingual calibration is not good for LLMs, unless dedicated calibration strategies, like TS, are used to improve them.

Despite the amazing progress of LLMs across several benchmarks, they unfortunately suffer from sometimes generating incorrect answers, with very high confidence. This can have severe consequences for safety-critical applications such as healthcare, autonomous driving, or finances where mistakes can be very costly. With the increasing application of LLMs to such tasks, it is crucial to understand whether the predictions are reliable, and when the models are unsure of the answer. Confidence calibration is one such reliability metric that measures whether the model's prediction probability estimates are aligned with the actual probability of the answer being correct. Confidence calibration has been studied in Computer Vision Guo et al. (2017); Minderer et al. (2021) and Natural Language Processing Desai & Durrett (2020); Dan & Roth (2021). However, most of the prior works are limited to classification settings, which is inapplicable to the generality of the QA task. Recently, Zhang et al. (2021); Jiang et al. (2021) have shown that state-of-the-art English QA models are surprisingly poorly calibrated. However, there remains a gap in understanding of the calibration properties of multilingual QA models. In this paper, we address this gap by a comprehensive study on the *Calibration of Multilingual Question Answering Large Language Models*. The main research questions we investigate in this paper are:

*1) How well are MLLMs calibrated in the cross-lingual transfer scenario?*

*2) How can we improve MLLMs' confidence calibration on multilingual QA datasets?*

The contributions of our work are as follows[1]:

• We provide the first comprehensive benchmarking of confidence calibration of multilingual QA models (architectures including *extractive models*: mBERT, XLM-R and *generative models*: mT5, mBART, and LLaMa2) over both low- and high-resource languages, in-distribution and out-of-distribution settings.
• We observe that the calibration performance on English is not transferable to other languages, across various datasets and architectures. Distance between the target languages and English, and the distribution of different languages at the pre-training stage, are all highly correlated with calibration performance, across the various model types.
• An investigation of various calibration strategies including post-hoc methods and regularization methods, aimed at enhancing cross-lingual calibration. Temperature scaling (optimized over a cross-lingual validation dataset) shows the most significant improvement even if the target language is absent in the validation data.
• We consider the In-Context Learning (ICL) scenario for LLMs such as LlaMa-2 and show that ICL improves both accuracy and calibration on multilingual QA tasks.
• We perform a suite of ablation experiments to study the role of example diversity, language diversity,

---

[1]Our code is available at https://github.com/yangy96/multiqa-calib

and model size on calibration, and to compare the calibration of MLLMs with that of their monolingual counterparts.

## 2 Related Work

### 2.1 Calibration of Large Language Models

Prior studies (Desai & Durrett, 2020; Dan & Roth, 2021; Xu & Zhang, 2023; He et al., 2021) have investigated the calibration of pre-trained LLMs on downstream tasks, primarily with a focus on monolingual English models and tasks, and primarily in the classification setting. Kuleshov & Liang (2015); Jagannatha & Yu (2020) study calibration for structured prediction, but not in the context of LLMs, Recently, Ahuja et al. (2022) studies calibration of multilingual pre-trained LLMs, specifically mBERT and XLM (Devlin et al., 2019; Conneau et al., 2019) on various downstream classification tasks including natural language inference and commonsense reasoning. They show that multilingual models are not well-calibrated in the classification setting, especially for low-resource languages. Jiang et al. (2022) also explores cross-lingual calibration performance for mBERT and XLM, comparing various post-hoc calibration methods on both structured and unstructured prediction tasks. In this work, we extend this line of work to calibration of MLLMs for QA, in both classification and generative settings, and to cross-lingual and distribution shift settings.

### 2.2 Calibration of models on Question Answering tasks

Recently, there has been growing interest in studying calibration of English QA models (Kamath et al., 2020; Zhang et al., 2021; Jiang et al., 2021; Si et al., 2022). Kamath et al. (2020) trains an extra calibrator of confidence scores to improve the calibration performance and examines the calibration performance on an out-of-domain (OOD) setting. They utilize the scores from the calibrator and uses it as a reranker to select the answers. Zhang et al. (2021) extends this work by adding the features of the context and back-translated context. Jiang et al. (2021) analyzes the calibration performance of generative language models, and find that the generative models on QA are not well-calibrated. Our work in contrast investigates the calibration of pre-trained *multilingual* LLMs (both extractive, with an encoder-only architecture, and generative, with an encoder-decoder or decoder-only architecture) on QA, and various techniques to improve calibration such as temperature scaling Guo et al. (2017), label smoothing Szegedy et al. (2016) and cross-lingual data augmentation.

## 3 Background

### 3.1 Question Answering

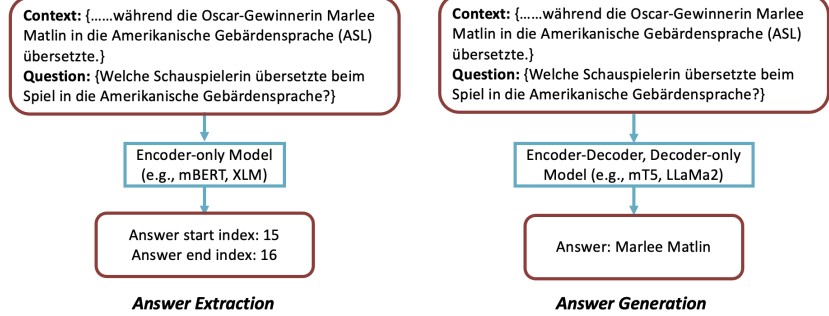

Figure 2: Differences in the output format between extractive and generative QA models.

In this work, we focus on two broad types of models for QA: extractive (or discriminative) and generative (as shown in Figure 2). For models based on encoder-only language models like mBERT and XLM-R (Conneau

et al., 2019; Devlin et al., 2019; Liu et al., 2019), the prediction of the answer span within the given context is framed as a classification task and achieved using two linear layers. These linear layers are placed on top of the hidden representations and are responsible for predicting the start and end indices of the answer span $Y$, respectively. The logit score of the answer $z_{ans}$ is defined as the sum of the logits of the start and end positions, $z_{start}$ and $z_{end}$ Si et al. (2022). Unlike standard classification problems which contain a fixed number of classes and compute the logits score from the model over those classes, here we sample a $K$-sized candidate set $\mathcal{I} = \{Y\}_K$ of answers for a question (Jiang et al., 2021), by selecting answers with top $K$ logits scores $z_{ans}$. The confidence of each candidate answer is the score after softmax is applied to its logit $z_{ans}$.

In the case of generative models like mT5 and mBART Xue et al. (2021); Liu et al. (2020); Touvron et al. (2023), we frame question-answering task as a sequence-to-sequence generation problem Khashabi et al. (2020). If $y_i$ denotes the $i^{th}$ generated token and $y_{<i}$ denotes all the previously generated $i - 1$ tokens, the probability of the answer is defined as $P'(Y|X) = \prod_i^{|Y|} P(y_i|X, y_{<i})$, a product of the individual token probabilities $P(y_i|X, y_{<i})$ that are generated by the decoder. Beam Search is used to find the candidate answers and the normalized probability is defined as $\hat{P}(\hat{Y}|X) = \frac{P_{LM}(\hat{Y}|X)}{\sum_{Y' \in \mathcal{I}} P'(Y'|X)}$, and used as the confidence of the prediction. The final answer of the model is the one with the highest normalized probability, over all answers in the candidate set.

## 3.2 Metrics to measure the Confidence Calibration of a model

A model is considered well-calibrated if the confidence estimates of its prediction are well-aligned with the actual probability of the answer being correct. Given an input $X$ for which the gold output is $Y$ and the model output is $\hat{Y}$, with confidence $\hat{P}$, a perfectly calibrated model satisfies the following condition:

$$P(\hat{Y} = Y \mid \hat{P}(\hat{Y} \mid X) = p) = p, \forall p \in [0, 1].$$

The above probability cannot be computed using finitely many samples since $\hat{P}$ is a continuous random variable and, in practice, it is approximated by bucketing the predictions.

**Expected calibration error (ECE)** A popular metric to measure confidence calibration is called the Expected Calibration Error (ECE) Guo et al. (2017). The predictions are bucketed into $M$ disjoint equally sized interval bins based on the confidence values, and the weighted average of the difference between each bucket's accuracy and confidence is calculated:

$$\sum_{m=1}^{M} \frac{|B_m|}{n} \mid acc(B_m) - conf(B_m) \mid,$$

where $B_m$ is the $m^{th}$ bucket containing samples whose prediction confidence lies in $(\frac{m-1}{M}, \frac{m}{M}]$, $acc(B_m)$ and $conf(B_m)$ is the average accuracy and prediction confidence of examples in the bucket $B_m$ respectively. We set the number of bins as 10 in all our experiments.

## 3.3 Techniques to improve the Confidence Calibration of a model

The calibration properties of a model can be evaluated directly out-of-box based on the probabilities it assigns to the predicted answers. Further, one can adopt strategies to calibrate a model, which can be broadly categorized into:
• Post-hoc calibration methods that do not require any additional training, for example, Temperature Scaling (TS) Guo et al. (2017).
• Specialized fine-tuning, such as Label Smoothing and Few Shot Learning, which regularizes training, or leverages augmented data respectively.

**Temperature Scaling (TS)** is a technique to improve the calibration of a machine learning model (Guo et al., 2017). When applying the softmax function to output the probability distribution of the logits, TS utilizes a single parameter $\tau$ to scale the logits: $softmax(z_i) = \frac{exp(z_i/\tau)}{\sum_i^K exp(z_i/\tau)}$. We perform TS in two ways specialized to the QA model type:

- *Extractive Models*: we compute the TS factors on the start logits and the end logits separately ($\tau_{start}$ and $\tau_{end}$), obtained by optimizing the Negative Log-Likelihood (NLL) loss on the validation set, extending the standard classification setting in Guo et al. (2017).The softmax score of the answer is computed as follows:

$$softmax(z_{ans}) = \frac{exp(\frac{z_{start}}{\tau_{start}} + \frac{z_{end}}{\tau_{end}})}{\sum_i^K exp(\frac{z_{start}}{\tau_{start}} + \frac{z_{end}}{\tau_{end}}))}$$

- *Generative Models*: For each of the $K$ candidate answers, we have an associated normalized probability $\hat{P}(\hat{Y} \mid X)$. We use the log probabilities of these as logits $z = log\hat{P}(\hat{Y} \mid X)$, in the softmax function Jiang et al. (2021). We then similarly optimize for a temperature $T$, with respect to the NLL loss, on the validation set.

Because minimizing cross-entropy during training encourages the model to make the correct-class logit arbitrarily large, it often produces overconfident predictions. Temperature scaling learns an optimal scalar $T$ that rescales these logits so that the predicted probabilities more closely match empirical accuracy, thereby improving calibration.

**Few-shot Learning (FS)** Incorporating a small number of examples from the target language during fine-tuning, mixed in with the English examples improves the calibration performance in classification tasks (Ahuja et al., 2022). Pre-trained language models are predominantly trained on English, which leads them to make overconfident predictions on non-English inputs, especially those from distant languages (Jiang et al., 2022). Incorporating a small set of cheaply translated multilingual examples during fine-tuning provides the model with direct exposure to these less-familiar languages, helping it better estimate its uncertainty and reducing the confidence bias when handling non-English text. We investigate whether this strategy helps calibrate multilingual QA models by randomly selecting 1000 samples from 5 different languages to fine-tune LLMs, in addition to the almost 100 times larger English data.

## 4 Benchmarking Calibration of Multilingual QA Models

In this section, we perform a comprehensive set of experiments spanning various LLMs and multi-lingual QA datasets. We focus on the *cross-lingual zero-shot* setting where the models are fine-tuned only on the English QA training data and tested on other languages not seen during fine-tuning, and also show results under challenging distribution shift scenarios.

**Pre-trained MLLMs:** In our experiments, we investigate the calibration performance of five different models: mBERT (Devlin et al., 2019), XLM-R (Conneau et al., 2019), mT5 (Xue et al., 2021), mBART (Liu et al., 2020), LLaMa2 (Touvron et al., 2023), and Aya-expanse (Dang et al., 2024). mBERT and XLM-R are encoder-only extractive architectures, while mBART and mT5 are encoder-decoder generative architectures. LLaMa2 ana Aya are decoder-only LLMs with much greater ICL capabilities compared to mT5/mBART, and we use `Llama-2-7b-hf` and `aya-expanse-8b`. We fine-tune all these models on the SQuAD 1.1 training data (Rajpurkar et al., 2016)[2].

**Datasets:**[3]
- XQuAD (Artetxe et al., 2019) is a popular benchmark to evaluate the cross-lingual ability of multilingual models on QA. The dataset is derived from the development set of SQuAD 1.1. It contains 1190 parallel pairs for 12 languages including English.
- MLQA (MultiLingual Question Answering) has a similar extractive-answer format as XQuAD and contains QA pairs in seven languages. (Lewis et al., 2019b)
- TyDiQA (TyDiQA-GoldP) (Clark et al., 2020) also shares the extractive-answer format , and contains 9 topologically diverse languages such as Korean and Telugu[4].

---

[2]The size of the candidate set, $K$, is set as 20 for all MLLMs and for LLaMa2, $K$ is set at 10

[3]Additional details of the datasets and experiments are presented in Appendix A.1.

[4]We use the secondary task (Gold-P) in TyDiQA, which aims at predicting the specific span within a context passage that serves as the answer to the question. The TyDiQA data is collected directly in each language without the use of translation (unlike MLQA and XQuAD).

**Metrics**
• EM: Exact Match Rate. The prediction should be the same as the gold answer.
• ECE: Expected Calibration Error. In this paper, ECE denotes the Expected Calibration Error averaged over all the languages in the corresponding dataset.
• ECE(en): ECE(en) denotes the Expected Calibration Error on the English-data only.

### 4.1 Results for In-Distribution Tasks

Figure 3 and Table 1 show the calibration performance of the five different MLLMs for the XQuAD dataset in the cross-lingual zero-shot setting. We notice that the relative increase in answer error for languages other than English is smaller compared to the relative increase in ECE across all models. For example, as shown in Table 1, the average prediction error for all the non-English languages is 56% while for English answer error is 33%, i.e. a 69.7 % increase. On the other hand, the average ECE for non-English languages is 18 % vs 7.32 % for English, an increase of 145%. LLaMa2 has a comparable calibration performance as the extractive models but the exact match rate is lower, especially for languages that are less represented in LLaMa2's pre-training data, such as Hindi and Greek[5].

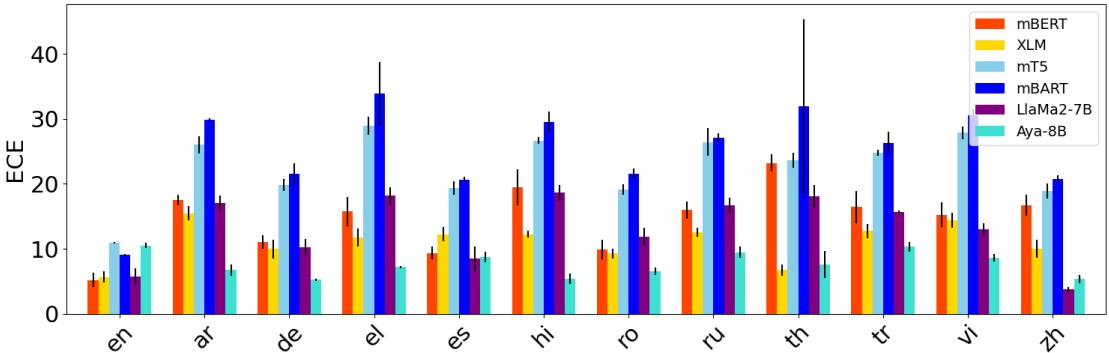

Figure 3: Calibration performance of six different models on the XQuAD dataset. Note that ECE is lower the better. mBART gets higher variance on TH and EL because it has not seen the two languages at the pre-training stage.

Table 1: Average performance across six different models on the XQuAD dataset (12 languages). EM is higher the better, ECE is lower the better.

| EN | | AR | | DE | | EL | | ES | | HI | |
|---|---|---|---|---|---|---|---|---|---|---|---|
| EM | ECE | EM | ECE | EM | ECE | EM | ECE | EM | ECE | EM | ECE |
| 70.24 | 7.84 | 42.37 | 18.78 | 55.47 | 13.0 | 39.38 | 19.27 | 55.20 | 13.13 | 43.63 | 18.66 |
| RO | | RU | | TH | | TR | | VI | | ZH | |
| EM | ECE | EM | ECE | EM | ECE | EM | ECE | EM | ECE | EM | ECE |
| 56.70 | 13.06 | 45.42 | 18.01 | 40.52 | 18.54 | 44.00 | 17.70 | 48.82 | 18.26 | 56.47 | 12.59 |

### 4.2 Results for Out-of-Distribution (OOD) Tasks

Additionally, we evaluate the performance of MLLMs on the TyDiQA dataset, which is collected from a different resource and different setup than SQuAD, resulting in a distribution shift for the model fine-tuned on SQuAD. In Table 2 we show the performance of the MLLMs on the TyDiQA test set, and observe the following:
•Fine-tuning on SQuAD helps more than fine-tuning on TyDiQA (en), even though the task is OOD. This can be attributed to the much larger amount of fine-tuning data in SQuAD compared to TyDiQA-en which

---

[5]The accuracy of the models on XQuAD are in Table 16 in Appendix. The language distribution of pre-training LLaMa2 is in Appendix A.6

makes up for the task distribution shift.

• Fine-tuning on all TyDiQA languages helps much more than just TyDiQA-en or SQuAD.

Table 2: Comparison of the calibration performance of various multilingual LLMs. The models are fine-tuned on SQuAD, TyDiQA-en, and TyDiQA-all, respectively and are evaluated on the test set of TyDiQA. Note that the results are averaged over all the 9 languages.

| XLM | EM(↑) | ECE(↓) | ECE(en)(↓) | mT5 | EM(↑) | ECE(↓) | ECE(en)(↓) |
|---|---|---|---|---|---|---|---|
| SQuAD | 50.04 | 14.78 | 14.11 | SQuAD | 50.31 | 24.37 | **18.47** |
| TyDiQA-en | 35.74 | 16.40 | **7.33** | TyDiQA-en | 40.13 | 26.96 | 19.16 |
| TyDiQA-all | **66.09** | **12.83** | 11.37 | TyDiQA-all | **67.11** | **18.03** | 19.22 |

In our experiments, for both in-domain and out-of-domain tasks, we find that extractive and generative models (scaled from 100M to 8B parameters) are consistently more miscalibrated on non-English inputs, with substantially higher calibration error than on English. This persistent cross-lingual calibration gap highlights the need to improve prediction reliability in multilingual settings.

## 5 Techniques to Improve the Calibration of MLLMs

In this section, we explore the role of various strategies to improve calibration including post-hoc methods, regularization methods, and ICL. We aim to address the following questions:

• Do existing calibration techniques work in our cross-lingual transfer settings?

• Can data-augmented fine-tuning on translated cross-lingual data improve calibration?

• What are the comparative impacts of having more monolingual data versus having more diverse, cross-lingual data?

• Can using in-context examples improve confidence calibration?

### 5.1 Post-hoc Methods: Temperature Scaling

In Tables 12 we demonstrate the benefits of using temperature scaling (TS) and few-shot learning (FS) on calibration for extractive models and generative models. TS does not affect accuracy by design, but it provides significant benefits in calibration in most cases. The optimized $T > 1$ pushes overconfident predictions into confidence ranges that more accurately reflect their true correctness, thereby reducing calibration error. Our experiments explore the impact of different validation sets: 1) the SQuAD validation dataset (10.6k English sentences); 2) Merged MLQA validation dataset (with 7 languages, 3k sentences). We also observe that optimizing the temperature on a relatively small multilingual validation dataset is more powerful than on a larger English-only validation dataset. Notably, even though some of the languages (e.g. SW, KO) do not occur in the merged validation dataset of XQuAD and TyDiQA, the temperature computed on the merged dataset still effectively improves the calibration performance in those languages[6].

Table 3: Calibration performance after applying temperature scaling (TS) and few-shot learning (FS) on mBERT and mT5, evaluated on the XQuAD test set.

| **mBERT** | EM(↑) | ECE(↓) | ECE(en)(↓) | **mT5** | EM(↑) | ECE(↓) | ECE(en)(↓) |
|---|---|---|---|---|---|---|---|
| | 46.77 | 16.36 | 6.78 | | 50.95 | 23.06 | 10.88 |
| TS (SQuAD) | 46.77 | 6.3 | 6.66 | TS (SQuAD) | 50.95 | 14.05 | 4.84 |
| TS (Merge) | 46.77 | **5.86** | 7.87 | TS (Merge) | 50.95 | **10.19** | **2.77** |
| FS | **48.38** | 5.87 | **5.27** | FS | **54.10** | 21.22 | 11.04 |

---

[6]Label Smoothing experiments and the observed benefits on calibration are in Appendix A.3.

## 5.2   Data Augmentation via Translation

We now investigate the effects of augmenting the training data by incorporating translated data. In our experiments, we sampled 9929 training examples in English and obtained their translations in five different languages $(AR, DE, ES, HI, VI)$[7]. We have four dataset configurations: **En** denotes a subset of the English data; **En-Large** denotes the full English data with available translations; **En-tr** denotes the En subset along with its translations in other languages; **Mixed** denotes each subset is from a different language. We fine-tune the MLLMs on this mixed dataset and evaluate their calibration performances. We show the detailed results in Table 4[8]. We see that data augmentation via translation improves calibration performance (even for languages not included in the translation) by almost 75% and we observe the following helpfulness ranking: Mixed > En-tr > En-large > En.

Table 4: Calibration after adding translated data during fine-tuning mBERT and mT5, evaluated on the XQuAD test set. The size of En, En-tr is 9929, and En-large, Mixed is 59574.

| mBERT | EM($\uparrow$) | ECE($\downarrow$) | ECE(en)($\downarrow$) | mT5 | EM($\uparrow$) | ECE($\downarrow$) | ECE(en)($\downarrow$) |
|---|---|---|---|---|---|---|---|
| En | 43.51 | 17.75 | 9.93 | En | 49.88 | 24.05 | 18.36 |
| En-tr | 46.69 | 5.58 | 10.91 | En-tr | 53.63 | 22.37 | 14.66 |
| En-large | 46.18 | 12.18 | **3.73** | En-large | 50.99 | 23.27 | **11.91** |
| Mixed | **49.39** | **5.56** | 10.47 | Mixed | **54.93** | **20.97** | 13.48 |

## 5.3   In-context Learning

In-context learning (ICL) (Brown et al., 2020) has been used as an efficient way to direct LLMs to quickly adapt to a new task. It appends the task demonstrations with the input prompts which enables the LLMs to learn the task from the examples effectively. While ICL is known to boost accuracy, here, we demonstrate it also improves calibration on multilingual QA[9]. We experimented with two ICL variants: 1) RANDOM: we randomly select two samples of the target language from the training dataset 2) ADAPTIVE: we choose the two samples of the target language from the training data most similar to the test input, measured by the similarity between the contextual embeddings. Specifically, we use the contextual embedding from the encoder LLM for the test input and all the training examples and find the examples with the highest cosine similarity.

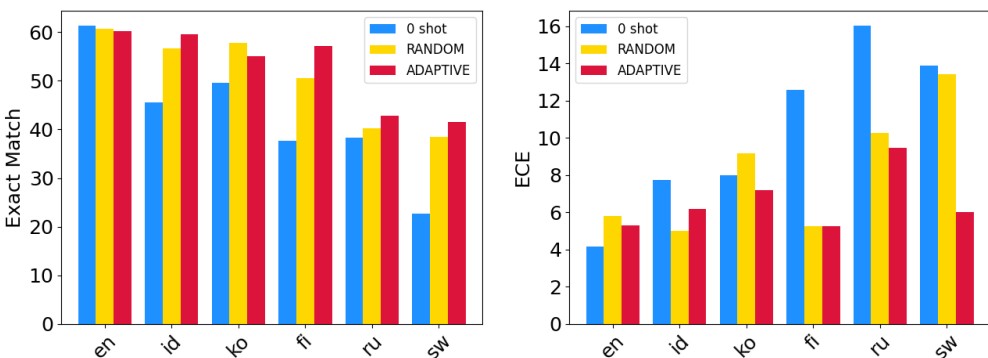

Figure 4: Comparison of accuracy (Left) and calibration (Right) of LLaMa2 across six languages of TyDiQA under 1) zero-shot setting, 2) RANDOM: random 2-shot ICL 3) ADAPTIVE: most similar 2-shot ICL. Lower ECE, and higher EM indicate better performance.

---

[7]The translated sentences are obtained from the MLQA-translated dataset.

[8]We diagrammatically illustrate the different configurations in Figure 10 in Appendix A.11

[9]ICL is particularly useful in calibrating massive LLMs such as LLaMa-2, where TS becomes prohibitively expensive due to temperature optimization on the validation data.

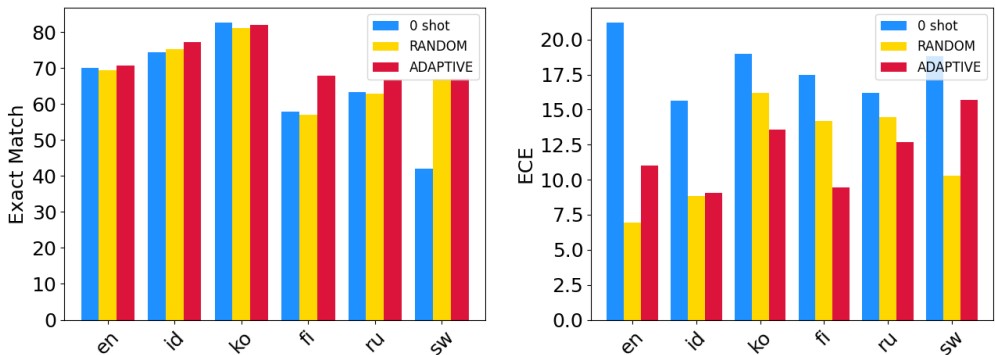

Figure 5: Comparison of accuracy (Left) and calibration (Right) of Aya across six languages of TyDiQA under 1) zero-shot setting, 2) RANDOM: random 2-shot ICL 3) ADAPTIVE: most similar 2-shot ICL. Lower ECE, and higher EM indicate better performance.

We present and compare the calibration performance of LLaMa2-7B (fine-tuned on SQuAD) and Aya-expanse-8B across six different languages with different ICL choices on TyDiQA in Figure 4 and Figure 5 [10]. We observe that the exact match rate is low for zero-shot learning but ICL with 2 samples can significantly improve the performance for Llama-2. For Aya, English has higher ECE than the other languages. Nevertheless, in-context learning still remains an efficient way to improve confidence calibration, consistent with our observations on Llama-2. We additionally demonstrate that selecting the most similar 2 samples for the in-context learning improves both accuracy and calibration performance, and decreases ECE by more than 10% compared to RANDOM ICL. For a low-resource language like Swahili, ADAPTIVE ICL generates a much better-calibrated output compared to zero-shot and RANDOM for LlaMa-2. Figure 6 shows some qualitative examples where ICL helps in model calibration for Korean and Swahili.

| | **Prediction** | | **Confidence** |
|---|---|---|---|
| **Context:** 1894년에 창단하였으며 2000년부터 현재까지 코메리카 파크를 홈구장으로 사용하고 있다. **Question:** 디트로이트 타이거스 창단일은 언제인가요? | w/o ICL → 1894년 | ✔ | 0.52 |
| | w ICL → 1894년 | ✔ | 0.72 |
| **Context:** Jean Bedel Bokassa (* 22 Februari 1921 - † 3 Novemba 1996) alikuwa rais na baadaye Kaisari wa Jamhuri ya Afrika ya Kati au baadaye Milki ya Afrika ya Kati hadi kupinduliwa tar. 21 Septemba 1979. **Question:** Je,Jean Bedel Bokassa alizaliwa mwaka upi? | w/o ICL → 21 Septemba 1979 | ✘ | 0.46 |
| | w ICL → 1979 | ✘ | 0.28 |

Figure 6: When appending the Korean and Swahili examples in the prompts, the model is more confident about the correct prediction and less confidence about the wrong prediction.

Given our experimental results, we confirm that existing calibration techniques are effective in cross-lingual settings. In particular, optimizing temperature on a relatively small but semantically diverse multilingual validation set yields stronger multilingual confidence calibration than temperature scaling on English only and other calibration techniques such as label smoothing and few-shot learning. We also observe that data augmenting with semantically diverse cross-lingual data is more helpful than simply adding more monolingual data for improving multilingual confidence calibration. The observations apply for both extractive and generative models. Finally, for more powerful decoder-only language models such as LLaMA-2-7B and Aya-8B, in-context learning improves both accuracy and confidence calibration across languages, with larger gains for lower-resource languages.

---

[10]The results of the random 2-shot ICL experiment are averaged over 5 runs.

# 6 Additional Ablation Experiments and Discussion

In this section, we perform ablations to answer the following research questions:
• the relationship between linguistic and non-linguistic features, and the transferability of calibration performance from English to other languages.
• the effect of model size on calibration, and accuracy, for a particular model family.[11]

## 6.1 Investigating the effects of language distance and corpus size on calibration

One important aspect of the multilingual model analysis is the factors that affect the model's cross-lingual ability (Jones et al., 2021; Jiang et al., 2022; Ahuja et al., 2022). Here we focus on analyzing both linguistic (distances between languages) and non-linguistic factors (pre-training data size) for calibration performance.
**Lingustic Features** Following previous work (Ahuja et al., 2022; Jones et al., 2021), we load the *syntactic* and *genetic* distances that are pre-computed by the *URIEL* project (Littell et al., 2017), as the distance measurements between English and other target languages. The syntactic distance measures the structural differences between various languages. The genetic distance quantifies the genetic relation between languages according to the Glottolog tree. We investigate whether the closeness of the target language to the source language, English, implies better calibration performance. To measure this we compute Pearson's correlation coefficient between the language distance and ECE of the standard models. Table 5 shows that the calibration performance is highly correlated with the syntactic distances between English and the corresponding languages.
**Non-Lingustic Features** We also explore the impact of non-linguistic features from training. In this section, we compute the correlation between pre-training data size and calibration performance and Table 5 indicates that the size of different languages in the pre-training influences the cross-lingual calibration of QA models.

Table 5: Pearson's correlation coefficient between linguistic/non-linguistic characteristics and calibration error on the XQuAD dataset for various models (Left: Extractive QA models, Right: Generative QA models). Higher absolute values indicate better correlation. *Syn* and *Gen* denote the syntactic and genetic distance between English and other languages respectively, and *Size* indicates the proportion of the pre-training data in each language.

| Model | Syn | Gen | Size | Model | Syn | Gen | Size |
|-------|-----|-----|------|-------|-----|-----|------|
| mBERT | **0.74** | 0.68 | -0.72 | mT5 | **0.77** | 0.69 | -0.68 |
| XLM-R | **0.56** | 0.54 | -0.40 | mBART | **0.76** | 0.72 | -0.31 |
|       |     |     |      | LLaMa2 | **0.60** | 0.53 | -0.57 |

## 6.2 Investigating the effects of model size on calibration

Lastly, we investigate the effect of model size on confidence calibration. While Guo et al. (2017) demonstrated that calibration degrades with increasing model size (on ResNet), we show that this is not the case for pre-trained MLLMs. We demonstrate the effect of model size on extractive and generative models for XQuAD and TyDiQA (Tables 6)[12]. The table shows that the accuracy increases with model size, as expected. Further, we note that confidence calibration also improves with increasing model size, which also supports prior findings on the calibration of monolingual encoder LLMs in Dan & Roth (2021).

In this section, we have examined how both linguistic and non-linguistic factors shape calibration performance. On the linguistic side, we focus on the syntactic and genetic distance between English and the target language; on the non-linguistic side, we consider pre-training size. We find that all these factors affect the model's calibration performance. In addition, we observe a consistent scaling pattern: as model size increases, accuracy improves and calibration error decreases.

---

[11]We additionally investigate how the calibration of multilingual models corresponds to their monolingual counterparts in Appendix A.13.2.
[12]Additional model details are in Appendix A.2.

Table 6: Calibration performance of XLM-R (Left) and mT5 (Right) across different sizes, evaluated on the XQuAD test set and TyDiQA test sets.

| XLM-R | XQuAD | | TyDiQA | | mT5 | XQuAD | | TyDiQA | |
|---|---|---|---|---|---|---|---|---|---|
| | EM($\uparrow$) | ECE($\downarrow$) | EM($\uparrow$) | ECE($\downarrow$) | | EM($\uparrow$) | ECE($\downarrow$) | EM($\uparrow$) | ECE($\downarrow$) |
| | | | | | **Small** | 38.44 | 27.44 | 32.05 | 30.29 |
| **Base** | 54.93 | 12.38 | 50.04 | 14.78 | **Base** | 50.95 | 23.06 | 50.31 | 24.37 |
| **Large** | **60.50** | **10.69** | **61.02** | **13.42** | **Large** | **56.02** | **21.60** | **56.91** | **20.27** |

## 7 Conclusion

In this work, we performed a comprehensive study of the confidence calibration of MLLMs, focusing on the QA task. We studied diverse scenarios, covering extractive and generative LLMs, several QA datasets, different calibration strategies, different inference settings (ICL and fine-tuning), and generalization under distribution shifts. We summarize the key insights from our paper: (1) Multilingual models need to be calibrated, especially in zero-shot settings, before deployment in real-world applications. 2) Temperature scaling on a mixed-language validation dataset is a very effective calibration strategy. 3) Adding cheap machine-translated data at the fine-tuning stage helps improve calibration even on languages unseen during fine-tuning. 4) ICL benefits not only the accuracy of powerful LLMs, but also their confidence calibration on multilingual tasks. We believe our work will be fundamental in spurring progress towards developing reliable multilingual systems and advancing NLP forlow-resourced languages.

## Acknowledgment

We thank the anonymous reviewers for their constructive feedback and suggestions. This research was sponsored by the Army Research Office and was accomplished under Grant Number W911NF-20-1-0080. The views expressed are those of the authors and do not reflect the official policy or position of the Army Research Office or the U.S. Government.

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

# A    Appendix

## A.1    Dataset Details

Here we provide more statistics of the datasets.

Table 7: The number of training and test samples in SQuAD, XQuAD, MLQA, TyDiQA-GoldP.

|  | Train | Test |
|---|---|---|
| SQuAD | 87.6K | 10.6K |
| XQuAD | N/A | 14.28K |
| MLQA | N/A | 46.14K |
| TyDiQA-GoldP | 49.88K | 5.08K |

Table 8: List of languages in XQuAD, MLQA, and TyDiQA-GoldP

| XQuAD | MLQA | TyDiQA-GoldP |
|---|---|---|
| Arabic | Arabic | Arabic |
| German | German | Bengali |
| Greek | English | Finnish |
| English | Spanish | English |
| Spanish | Hindi | Swahili |
| Hindi | Vietnamese | Korean |
| Romanian | Chinese | Indonesian |
| Russian |  | Russian |
| Thai |  | Telugu |
| Turkish |  |  |
| Vietnamese |  |  |
| Chinese |  |  |

## A.2 Architecture Details

In section 6.2, we investigate the relationship between calibration performance and model size. Here, we provide more details of XLM-R and mT5 in a range of different sizes. XLM-R-base contains 12 layers with 768 hidden states and 8 heads while the large version contains 24 layers, 1027 hidden states, and 16 heads (Conneau et al., 2019; Wolf et al., 2020). The small variant of mT5 consists of 6 layers for both the encoder and decoder components, where the base mT5 model has 12 layers, and the large mT5 model has 24 layers.

Table 9: The number of parameters for XLM-R and mT5.

| No. of parameters | XLM-R | mT5 |
|---|---|---|
| **Small** | N/A | 300M |
| **Base** | 125M | 580M |
| **Large** | 355M | 1.2B |

## A.3 Experiment Details

The monolingual models in other languages are (section 6.2):

German-BERT: `https://www.deepset.ai/german-bert`

Arabic-BERT: `https://huggingface.co/asafaya/bert-base-arabic`

Chinese-BERT: `https://huggingface.co/bert-base-chinese`

All the experiments are run on two 48G NVIDIA RTX A6000 GPUs. During the fine-tuning process of the multilingual language models, a learning rate of 3e-5 is chosen from a hyperparameter selection range of [1e-5, 2e-5, 3e-5]. The batch size is set to 32. The extractive QA models are fine-tuned for 3 epochs and the generative QA models are fine-tuned for 5 epochs with AdamW optimizer. LLaMa2 is fine-tuned on SQuAD dataset for 1 epoch with QLoRA (Dettmers et al., 2023) mechanism (Smith, 2023) and Aya is fine-tuned for 1000 steps. It takes about 2 hours to train extractive models, about 7 hours to train generative models, and 24 hours to train LLaMa2. We follow the implementations from Huggingface (Wolf et al., 2020) to load the dataset and fine-tune the pre-trained models. The license of LLaMa2 can be found in `https://ai.meta.com/llama/license/`, mT5 and multilingual BERT are using apache-2.0 license, mBART and XLM-R are using MIT license. TyDiQA is using apache-2.0 license, MLQA and XQuAD are using cc-by-sa-3.0. The models and datasets are consistent with their use in research.

## A.4 LLaMa2 experiments configurations

We provide additional details of in-context learning experiments in this section. The prompts used for querying LLaMa2 is

```
Extract the minimal span word from the following context that best
answers the question.
### Question:
{question}
### Context:
{context}
### Answer:
```

The encoder-only language model used for extracting contextual embedding is sentence-transformers/stsb-xlm-r-multilingual Reimers & Gurevych (2019).

## A.5 Additional calibration metrics

**Reliability Diagrams** is a visual depiction of confidence calibration. These diagrams plot the average accuracy of each bin $B_m$ ($Y$-axis), $acc(B_m)$ as a function of the bin confidences $conf(B_m)$ ($X$-axis, sorted

in increasing order). If the model is perfectly calibrated, then the diagram should plot the identity function ($Y = X$ line). The greater the deviation from the diagonal, the more miscalibrated the model is. We show an example of a reliability diagram in Figure 1.

### A.6 Language Distribution for different models

In this section, we show the language code mapping and the proportion of training data that comes from a specific language for various pre-trained multilingual LLMs.

Table 10: Language code mapping.

| LANG CODE | |
|---|---|
| EN | English |
| AR | Arabic |
| DE | German |
| EL | Greek |
| ES | Spanish |
| HI | Hindi |
| RO | Romainian |
| RU | Russian |
| TH | Thai |
| TR | Turkish |
| VI | Vietnamese |
| ZH | Chinese |
| KO | Korean |
| FI | Finnish |
| SW | Swahili |
| ID | Indonesian |
| BN | Bengali |
| TE | Telugu |

Table 11: The distribution of different languages in the pre-training data for LLaMa2, mT5, mBRART, mBERT, XLM-R, shown in percentages. Note that we only display the languages tested in the XQuAD and TyDiQa dataset.

| LANG | LLaMa2 | mT5 | mBART | mBERT | XLM-R |
|---|---|---|---|---|---|
| EN | 89.70 | 5.67 | 21.67 | 22.54 | 12.56 |
| AR | ≤0.005 | 1.66 | 2.04 | 0.68 | 1.17 |
| DE | 0.17 | 3.05 | 4.86 | 7.10 | 2.78 |
| EL | ≤0.005 | 1.54 | 0.0 | 0.33 | 1.96 |
| ES | 0.13 | 3.09 | 3.89 | 3.53 | 2.22 |
| HI | ≤0.005 | 1.21 | 1.47 | 0.19 | 0.84 |
| RO | 0.03 | 1.58 | 4.48 | 2.80 | 2.56 |
| RU | 0.13 | 1.21 | 20.30 | 2.80 | 11.61 |
| TH | ≤0.005 | 1.14 | 0.0 | 0.34 | 2.99 |
| TR | ≤0.005 | 1.80 | 1.52 | 1.03 | 0.87 |
| VI | 0.08 | 1.87 | 10.02 | 0.51 | 5.73 |
| ZH | 0.13 | 1.67 | 3.42 | 1.80 | 1.69 |
| KO | 0.06 | 1.14 | 3.96 | 0.66 | 2.26 |
| FI | 0.03 | 1.35 | 3.96 | 1.58 | 2.27 |
| SW | ≤0.005 | 0.5 | 0.0 | 0.07 | 0.07 |
| ID | 0.03 | 1.80 | 0.0 | 0.80 | 0.80 |
| BN | ≤0.005 | 0.91 | 0.0 | 0.16 | 0.35 |
| TE | ≤0.005 | 0.52 | 0.0 | 0.36 | 0.20 |

### A.7 Additional Calibration Result

#### A.7.1 Label Smoothing (LS)

(Szegedy et al., 2016; Muller et al., 2019) is a regularization technique that constructs a new target vector ($h_i^{LS}$) from the one-hot target vector ($h_i$), where $h_i^{LS} = (1 - \alpha)h_i + \alpha/C$ for a $C$ class classification problem. The range of the label smoothing weight $\alpha$ is from 0 to 1. For question answering via encoder-based models, $C$ corresponds to the length of the context, since we want to select one of $C$ start positions and one of $C$ end positions for the answer. Accordingly, the $\alpha$ mass for the correct index is distributed to all other indices. Note that we have two hyper-parameters $\alpha_1$ and $\alpha_2$ corresponding to the start and end locations, which can be selected independently via optimizing on the validation set. For all the Label Smoothing experiments, we set the hyperparameter $\alpha = 0.1$.

We show additional calibration performance after confidence calibration for mBERT, XLM-R and mBART on XQUAD.

Table 12: Calibration performance after applying temperature scaling (TS), label smoothing (LS), and few-shot learning (FS) on mBERT and mT5, evaluated on the XQuAD test set.

| mBERT | EM($\uparrow$) | ECE($\downarrow$) | ECE(en)($\downarrow$) | XLM-R | EM($\uparrow$) | ECE($\downarrow$) | ECE(en)($\downarrow$) |
|---|---|---|---|---|---|---|---|
| | 46.77 | 16.36 | 6.78 | | 54.93 | 12.38 | 6.84 |
| TS (SQuAD) | 46.77 | 6.3 | 6.66 | TS (SQuAD) | 54.93 | **4.16** | 4.25 |
| TS (Merge) | 46.77 | **5.86** | 7.87 | TS (Merge) | 54.93 | 4.29 | 5.86 |
| LS | 47.80 | 14.65 | **1.88** | LS | **55.08** | 9.88 | **4.23** |
| FS | **48.38** | 5.87 | 5.27 | FS | 53.99 | 4.39 | 5.16 |

Table 13: Calibration performance after applying temperature scaling (TS), label smoothing (LS) and few-shot learning (FS) on QA models: XLM-R and mBART, evaluated on the XQuAD test set.

| mBART | EM($\uparrow$) | ECE($\downarrow$) | ECE(en)($\downarrow$) |
|---|---|---|---|
| | 39.18 | 22.64 | 9.19 |
| TS (SQuAD) | 39.18 | 15.05 | 4.26 |
| TS (Merge) | 39.18 | **10.23** | **2.97** |
| FS | **54.14** | 17.49 | 9.7 |

### A.8 Additional Results on MLQA

Table 14 shows the accuracy and calibration performance (ECE) for mBERT, XLM-R, mT5, mBART, LLaMa2 on MLQA.

Table 14: Calibration performance of four different models on the MLQA dataset (7 languages). EM denotes exact match rate. $\uparrow$ ($\downarrow$) denotes higher (lower) is better respectively.

| | EN | | AR | | DE | | ES | | HI | | VI | | ZH | |
|---|---|---|---|---|---|---|---|---|---|---|---|---|---|---|
| | EM ($\uparrow$) | ECE ($\downarrow$) | EM ($\uparrow$) | ECE ($\downarrow$) | EM ($\uparrow$) | ECE ($\downarrow$) | EM ($\uparrow$) | ECE ($\downarrow$) | EM ($\uparrow$) | ECE ($\downarrow$) | EM ($\uparrow$) | ECE ($\downarrow$) | EM ($\uparrow$) | ECE ($\downarrow$) |
| mBERT | 58.28 | 12.95 | 24.76 | 36.08 | 32.42 | 33.62 | 33.0 | 36.21 | 28.99 | 30.83 | 29.16 | 33.50 | 36.31 | 25.79 |
| XLM-R | 57.06 | 12.79 | 28.82 | 30.29 | 31.84 | 30.31 | 36.20 | 31.64 | 42.21 | 18.24 | 36.59 | 27.79 | 40.28 | 20.58 |
| mT5 | 64.94 | 15.24 | 30.46 | 37.73 | 44.28 | 30.17 | 43.35 | 30.66 | 35.60 | 32.09 | 36.89 | 33.77 | 37.03 | 27.89 |
| mBART | 60.02 | 13.82 | 19.01 | 33.56 | 36.64 | 32.02 | 29.02 | 39.47 | 24.30 | 31.25 | 31.44 | 33.34 | 30.11 | 28.42 |
| LLaMa2 | 57.72 | 1.96 | 18.11 | 24.73 | 35.09 | 16.54 | 33.79 | 17.7 | 19.24 | 22.27 | 28.37 | 17.98 | 34.3 | 18.11 |

### A.9 Additional Results on TyDiQA

We present accuracy (Figure 7) and ECE (Figure 8) for XLM-R, mT5 and mBART across 9 languages on TyDiQA. We also show additional results of Out-of-Distribution comparison in Table 15.

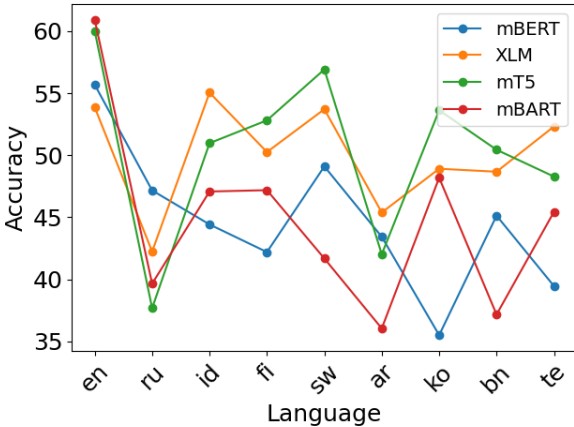

Figure 7: This figure compares the accuracy (EM) of mBERT, XLM-R, mT5, mBART across nine languages of TyDiQA.

Table 15: Comparison of the calibration performance of mBERT and mBART. The models are evaluated on the test set of TyDiQA. EM: exact match rate.

|  | EM($\uparrow$) | ECE($\downarrow$) | ECE(en)($\downarrow$) |
|---|---|---|---|
| mBERT-SQuAD | 44.68 | 17.30 | 11.46 |
| mBERT-TyDiQA-en | 44.99 | 11.12 | **5.95** |
| mBERT-TyDiQA-all | **68.74** | **10.98** | 8.02 |
| mBART-SQuAD | 37.41 | 27.04 | 16.56 |
| mBART-TyDiQA-en | 41.52 | 35.55 | 25.18 |
| mBART-TyDiQA-all | **64.87** | **14.33** | **12.35** |

### A.10 Additional Result on XQuAD

We show the detailed performance of five different models on XQuAD in Table 16.

We also show the individual ECE before/after calibration for mBERT, XLM-R, mT5 and mBART on XQuAD in Figure 9.

### A.11 Additional Results for Data Augmentation

Here we show a diagram for visualizing the setting of our data augmentation experiment in Figure 10 and provide additional results for XLM and mBART in Table 17.

Furthermore, it is worth noting that even for languages such as *RU* (Russian), *Ro* (Romanian) and *Tr* (Turkish), that were not included in the fine-tuning stage, we observed a significant improvement in their performance as shown in Figure 11. This suggests that the benefits of the fine-tuning process extend beyond the explicitly included languages and have a positive impact on the overall performance across various languages in the evaluation set. Theoretically studying how translated sentences affect fine-tuning performance, and the relation with language similarities is an interesting avenue for future study.

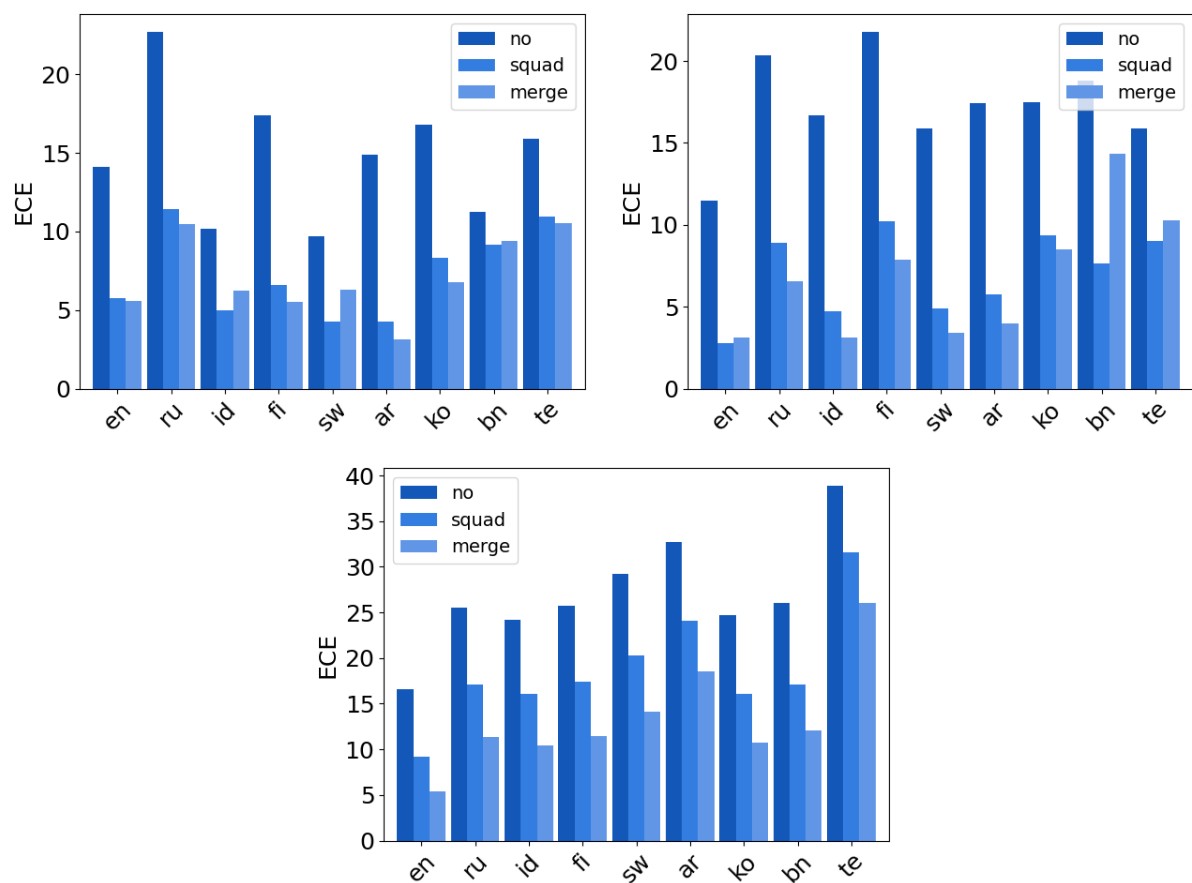

Figure 8: This figure compares the calibration performance (ECE) of XLM-R, mBERT, mBART across nine languages of TyDiQA under 1) no calibration 2) temperature scaling on English validation dataset 3) temperature scaling on a 7-language validation dataset.

Table 16: Calibration performance of four different models on the XQuAD dataset (12 languages). EM denotes exact match rate. ↑ (↓) denotes higher (lower) is better respectively. *Note that mBART has not seen EL even during pre-training so cannot generate answers correctly in that language.

| | EN | | AR | | DE | | EL | | ES | | HI | |
|---|---|---|---|---|---|---|---|---|---|---|---|---|
| | EM(↑) | ECE(↓) | EM(↑) | ECE(↓) | EM(↑) | ECE(↓) | EM(↑) | ECE(↓) | EM(↑) | ECE(↓) | EM(↑) | ECE(↓) |
| mBERT | 66.92 | 5.21 | 41.76 | 17.49 | 54.99 | 11.09 | 42.44 | 15.77 | 55.71 | 9.37 | 39.86 | 19.46 |
| XLM-R | 66.58 | 5.68 | 47.7 | 15.49 | 57.2 | 9.99 | 52.58 | 11.76 | 54.29 | 12.24 | 50.78 | 12.25 |
| mT5-R | 72.21 | 10.9 | 45.88 | 26.01 | 56.13 | 19.84 | 37.87 | 28.91 | 56.81 | 19.36 | 46.64 | 26.65 |
| mBART | 67.59 | 9.07 | 23.98 | 29.91 | 46.13 | 21.53 | 7.23* | 33.84* | 43.61 | 20.58 | 30.06 | 29.56 |
| LLaMa2 | 64.31 | 5.72 | 25.97 | 17.06 | 44.76 | 10.3 | 23.84 | 18.15 | 44.96 | 8.47 | 23.81 | 18.62 |
| | RO | | RU | | TH | | TR | | VI | | ZH | |
| | EM(↑) | ECE(↓) | EM(↑) | ECE(↓) | EM(↑) | ECE(↓) | EM(↑) | ECE(↓) | EM(↑) | ECE(↓) | EM(↑) | ECE(↓) |
| mBERT | 58.74 | 9.91 | 50.78 | 15.96 | 28.21 | 23.2 | 35.1 | 16.45 | 47.48 | 15.22 | 46.64 | 16.72 |
| XLM | 61.4 | 9.29 | 54.59 | 12.51 | 55.57 | 6.75 | 50.5 | 12.73 | 51.82 | 14.37 | 55.55 | 10.05 |
| mT5 | 59.19 | 19.11 | 39.55 | 26.43 | 49.92 | 23.62 | 49.08 | 24.79 | 47.76 | 27.84 | 57.93 | 18.92 |
| mBART | 47.79 | 21.57 | 32.3 | 27.04 | 22.86 | 31.89 | 40.08 | 26.32 | 36.69 | 30.52 | 47.9 | 20.73 |
| LLaMa2 | 36.41 | 11.91 | 33.61 | 16.68 | 22.49 | 18.13 | 23.89 | 15.62 | 37.06 | 12.99 | 57.23 | 3.73 |

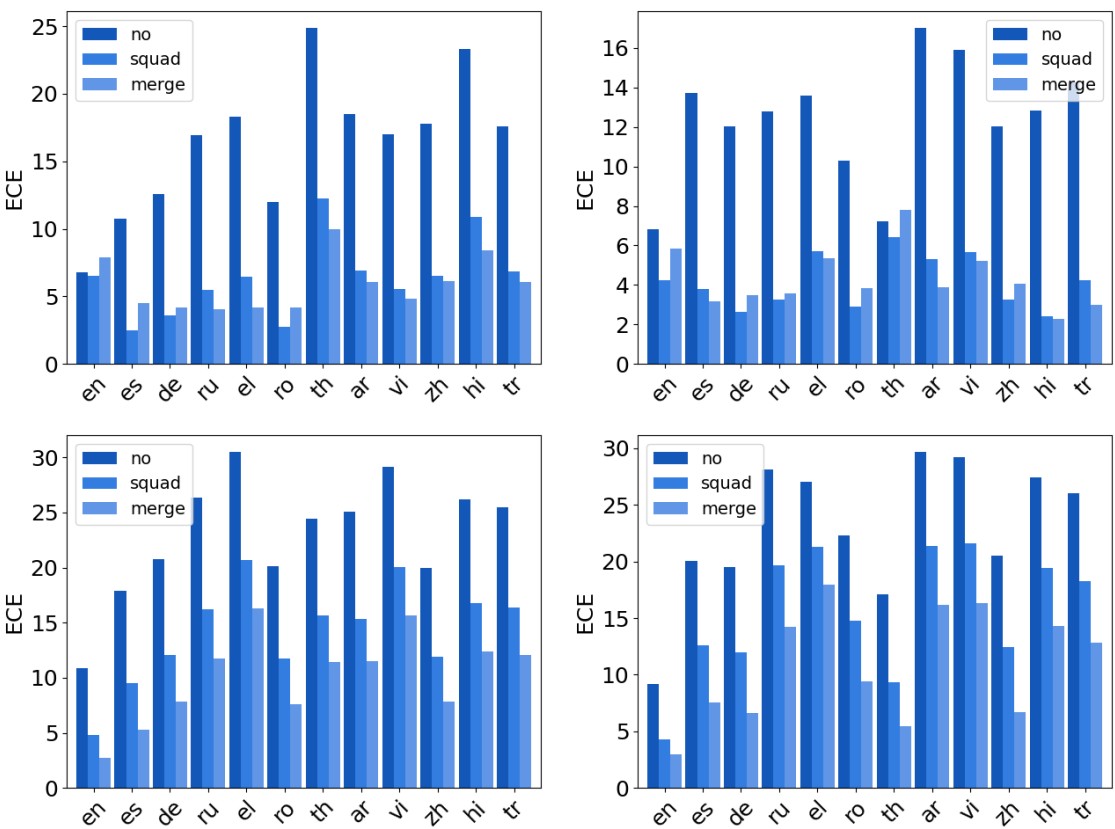

Figure 9: The figure presents results of the calibration performance, measured by ECE of various models - mBERT, XLM-R, mT5, and mBART - across 13 languages in the XQuAD dataset. 1) no calibration 2) temperature scaling on English validation dataset 3) temperature scaling on a 7-language validation dataset. ECE is lower the better.

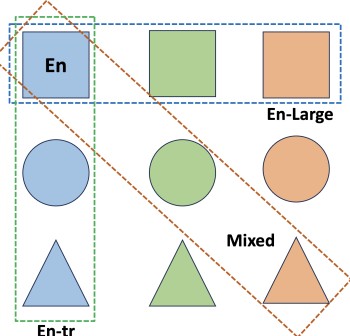

Figure 10: This figure describes the setting of Sec. 5.2. Different colors denote different examples and the different shapes denote different languages, eg: squares are English and circles are German. Thus each example (color) has a corresponding translation in the other languages (shapes). **En** denotes a subset of the English data and **En-Large** denotes the full English data with available translations. **En-tr** denotes the En subset along with its translations in other languages. **Mixed** denotes each subset from a different language. Note: Each colored shape has the same number of examples and thus En-Large, En-tr and Mixed have the same size.

Table 17: Calibration performance after adding translated pairs at the fine-tuning stage for XLM-R and mBART, evaluated on the XQuAD test set. In this case, the size of En is 9929, the size of the En-tr, and En-large, Mixed is 59574.

|  |  | EM($\uparrow$) | ECE($\downarrow$) | ECE(en)($\downarrow$) |
|---|---|---|---|---|
| XLM-R | En | 46.0 | 13.27 | 7.56 |
|  | En-tr | 49.65 | **5.24** | 11.34 |
|  | En-large | **53.66** | 11.82 | **5.82** |
|  | Mixed | 52.93 | 7.04 | 7.48 |
| mBART | En | 48.41 | 32.97 | 22.62 |
|  | En-tr | 49.96 | 32.42 | 24.02 |
|  | En-large | 45.18 | 35.79 | 20.83 |
|  | Mixed | **52.60** | **29.85** | **20.80** |

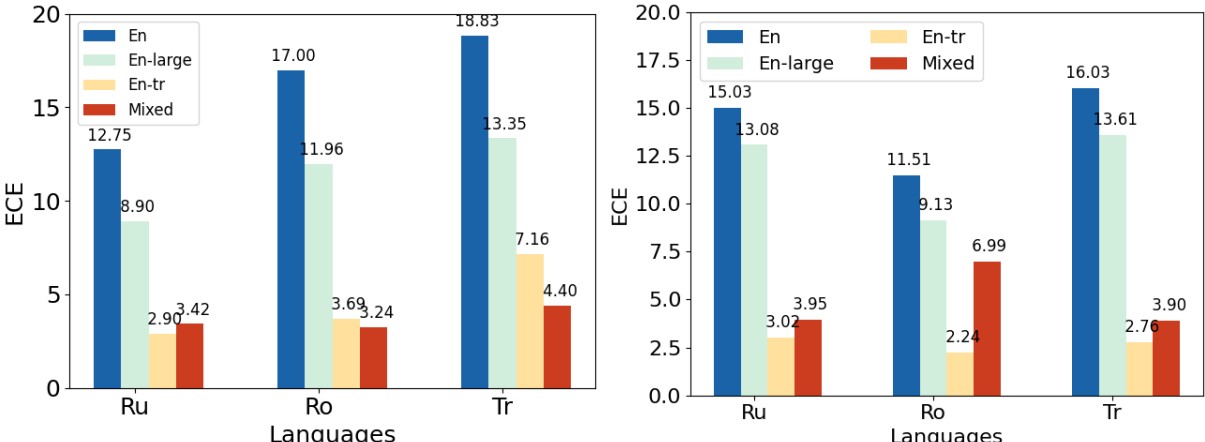

Figure 11: This figure shows the improvement of calibration brought by data augmentation via translation for different languages on mBERT (Left) and XLM-R (Right). **En** denotes a subset of the English data and **En-Large** denotes the full English data with available translations. **En-tr** denotes the En subset along with its translations in other languages. **Mixed** denotes each subset from a different language.

## A.12 Additional In-context Learning Results

Here we report additional results of confidence calibration with in-context learning. The model we used in this experiment is a chat-optimized decoder-only model, LlaMa2-7B-chat. Note that we report Validation Answer (VA), which checks whether the gold answer is contained in the generated answer, instead of Exact Match (EM).

Table 18: Calibration performance of LlaMa2-chat with/without ADAPTIVE ICL.

|  | EN | | ID | | FI | |
|---|---|---|---|---|---|---|
|  | VA | ECE | VA | ECE | VA | ECE |
| 0 | 41.36 | 10.82 | 26.19 | 14.16 | 18.29 | 23.18 |
| ADAPTIVE | 56.82 | 16.04 | 71.33 | 8.02 | 51.28 | 14.14 |
|  | KO | | SW | | RU | |
|  | VA | ECE | VA | ECE | VA | ECE |
| 0 | 38.41 | 14.53 | 7.01 | 45.57 | 27.09 | 12.41 |
| ADAPTIVE | 52.54 | 8.32 | 44.69 | 27.71 | 43.72 | 23.97 |

### A.13 Additional Results for Analysis

#### A.13.1 Linguistic distances versus calibration performance

We performed additional statistical testing to determine which factor significantly impacts calibration performance. More specifically, we applied the Wilcoxon signed-rank test as suggested in Dror et al. (2018) and compared the Pearson correlation coefficients for the three factors across different models and multiple runs. Our results indicate that the syntactic distance has more impact on the multilingual calibration performance compared to the genetic distance and pre-training size (with p-value= 0.0008 and p-value=0.0001 when alpha = 0.05). We also compute the correlation between other types of linguistic features in Table 19 and the calibration performance. More details about the features can be found in Littell et al. (2017).

Table 19: Pearson Correlation Coefficient between other linguistic characteristics and calibration error on XQuAD dataset for various models. The absolute value higher, the better. Here, Gen denotes the genetic distances between English and other languages, Geo indicates the geographical distance, Ive is the inventory distance, Pho denotes the phonological distance.

| Model | Gen | Geo | Ive | Pho |
|---|---|---|---|---|
| mBERT | 0.602 | 0.503 | 0.736 | 0.494 |
| XLM-R | 0.541 | 0.334 | 0.680 | 0.528 |
| mT5 | 0.687 | 0.213 | 0.629 | 0.353 |
| mBART | 0.716 | 0.301 | 0.751 | 0.495 |

XQuAD is a parallel dataset in 12 languages and another interesting question for the multilingual language model is: if a model is confident in predicting the answer of an English question-context pair, will it be confident in predicting the answer of a parallel question-context pair in other languages? Here, we compute the Pearson's correlation coefficient between the confidence for predicting answers of source language and those of target languages and plot the comparison in Figure 12. We observe that languages more similar to English exhibit a stronger correlation in the model's confidence levels for identical question-context pairs across languages. More specifically, when the model shows higher confidence in an English input, it tends to also display similar confidence in inputs from closer languages like Spanish and German. The correlations of the encoder-only models are much higher than encoder-decoder models.

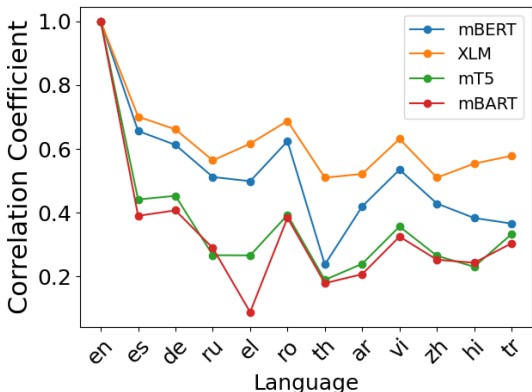

Figure 12: This figure presents the distance between English and other languages versus the correlation of the confidence between source language across four different models. Here Correlation Coefficient denotes Pearson's correlation coefficients. The higher the better. In the y-axis labels, languages are ordered by their syntactic distance from English.

#### A.13.2 Monolingual vs Multilingual Models

We also want to compare the multilingual models with their monolingual counterparts on extractive English QA, focusing on the SQuAD dataset. In Table 20, we compare the accuracy and ECE of English LLMs

and multilingual LLMs across both the extractive and generative architectures on the SQuAD test set. Our results confirm that multilingual models are generally worse than monolingual English LLMs when tested on English QA for either architecture type, which has also been noted in Wu & Dredze (2020). However, we notice a different trend for calibration. Both accuracy and confidence calibration are comparable when evaluating the multilingual model and English monolingual model on the same English task (both generative and extractive).

Table 20: Comparison of the calibration performance between English LMs and multilingual LMs. The models are evaluated on the test set of SQuAD.

|  | Exact Match (↑) | ECE (↓) |  | Exact Match (↑) | ECE (↓) |
|---|---|---|---|---|---|
| BERT | 76.61 | **4.09** | T5 | **80.67** | 5.42 |
| mBERT | **77.02** | 6.45 | mT5 | 78.90 | **5.32** |
| RoBERTa | **82.26** | **6.46** | BART | **84.11** | 29.24 |
| XLM-R | 77.21 | 8.46 | mBART | 77.14 | **8.05** |

Then we want to compare the multilingual models with their monolingual counterparts for languages other than English. We select BERT-German and BERT-Arabic (Safaya et al., 2020), and BERT-Chinese [13] and fine-tune them on the translated training dataset of SQuAD 1.1. The results are shown in Table 21 and Table **??**. We observe that the monolingual BERT models always achieve better calibration than their multilingual counterparts, although the multilingual models sometimes are more accurate than the corresponding monolingual models.

Table 21: Comparison of the calibration performance between German (DE)/Arabic (AR)/Chinese (ZH) BERT and multilingual BERT (mBERT). The models are evaluated on the XQuAD test set for each language.

|  | EM(↑) | ECE(↓) |  | EM(↑) | ECE (↓) |  | EM(↑) | ECE (↓) |
|---|---|---|---|---|---|---|---|---|
| BERT-DE | 52.10 | **4.71** | BERT-AR | **48.66** | **11.30** | BERT-ZH | 37.22 | **6.88** |
| mBERT | **54.37** | 12.55 | mBERT | 40.92 | 18.50 | mBERT | **46.81** | 17.78 |

Table 22: Comparison of the calibration performance between German (DE)/Arabic (AR)/Chinese (ZH) BERT and multilingual BERT (mBERT). The models are evaluated on the MLQA test set for each language.

|  | EM(↑) | ECE(↓) |  | EM(↑) | ECE (↓) |  | EM(↑) | ECE (↓) |
|---|---|---|---|---|---|---|---|---|
| BERT-DE | 28.13 | **17.96** | BERT-AR | **24.95** | **12.80** | BERT-ZH | 30.75 | **6.22** |
| mBERT | **32.42** | 33.62 | mBERT | 28.82 | 30.29 | mBERT | **36.31** | 25.79 |

### A.13.3 Size of Multilingual Models

Table 23: Calibration performance of mT5 and XLM-R across different sizes on the MLQA test sets.

|  | mT5 | | XLM-R | |
|---|---|---|---|---|
|  | EM(↑) | ECE(↓) | EM(↑) | ECE(↓) |
| **Small** | 28.88 | 33.80 | N/A | N/A |
| **Base** | 41.79 | 29.65 | 39.00 | **24.95** |
| **Large** | **45.73** | **27.40** | **43.94** | 26.86 |

---

[13]The details of the monolingual models of other languages are in Appendix A.3

