# OpenReview forum: "On Calibration of Multilingual Question Answering LLMs"
_TMLR — Accepted by TMLR_

### Review · Reviewer_UJ5S · 2025-08-01

**Summary Of Contributions:**

The paper performs a comprehensive empirical evaluation of the confidence calibration on Multilingual Large Language Models.  The paper found that the MLLMs need to be calibrated specially in zero-shot, temperature scaling is effective specially on the mixed-language validation setup, adding data in the fine-tuning step helps to improve the calibration, and that in-context learning improves the accuracy and confidence calibration on multilingual tasks.

**Audience:**

Yes

**Audience Explanation:**

Yes, the LLM community will be interested in this empirical findings.

**Broader Impact Concerns:**

I do not have any concerns.

**Claims And Evidence:**

Yes

**Claims Explanation:**

The paper presents a comprehensive set of results that support the claims of the results.  The descriptions and interpretations of the results could be better.

**Requested Changes:**

Overall, the paper presents a comprehensive set of results.  Each section links the results and presents the setup.  However, the final conclusions and claims could be better presented and expanded upon.  Perhaps, highlighting them or reiterating the main findings at the end of each (sub)section could help the reader to better understand the paper and get back to particular results more easily.

---

> ### Author Response · Authors · 2025-09-06
> **Response to Reviewer UJ5S**
>
> We appreciate the reviewer for your positive review and suggestions. Here we provide  take-away messages for each section and we also have updated them in the manuscript.
>
> **Section 4**:\
> For both in-domain and out-of-domain tasks, we find that extractive and generative models (scaled from ~100M to 8B parameters) are consistently more miscalibrated on non-English inputs, with substantially higher calibration error than on English. This persistent cross-lingual calibration gap highlights the need to improve prediction reliability in multilingual settings.
>
> **Section 5**:\
> Given our experimental results, we confirm that existing calibration techniques are effective in cross-lingual settings. In particular, optimizing temperature on a relatively small but semantically diverse multilingual validation set yields stronger multilingual confidence calibration than temperature scaling on English only and other calibration techniques such as label smoothing and few-shot learning. We also observe that data augmenting with semantically diverse cross-lingual data is more helpful than simply adding more monolingual data for improving multilingual confidence calibration. The observations apply for both extractive and generative models. Finally, for more powerful decoder-only language models such as LLaMA-2-7B and Aya-8B, in-context learning improves both accuracy and confidence calibration across languages, with larger gains for lower-resource languages.
>
>
> **Section 6**:\
> In this section, we have examined how both linguistic and non-linguistic factors shape calibration performance. On the linguistic side, we focus on the syntactic and genetic distance between English and the target language; on the non-linguistic side, we consider pre-training size. We find that all these factors affect the model’s calibration performance. In addition, we observe a consistent scaling pattern: as model size increases, accuracy improves and calibration error decreases.

---

> > ### Comment · Reviewer_UJ5S · 2025-09-08
> >
> > I thank the authors for the response to my concerns.  While they are not major, I think these changes will improve the readability of the results

---

### Review · Reviewer_VPdc · 2025-08-16

**Summary Of Contributions:**

This paper proposes a benchmark for calibration multilingual question answering. They want to show that the current LLM is not well-calibrated for other languages other than English.

**Additional Comments:**

NA

**Audience:**

Yes

**Audience Explanation:**

Some individual interest in model calibration might be of interest to this paper.

**Claims And Evidence:**

Yes

**Claims Explanation:**

The main claim of this paper is  "multilingual QA models are poorly calibrated for languages other than English" and " incorporating a small set of cheaply translated multilingual samples during fine-tuning/calibration effectively enhances the calibration performance".


For the first claim:

- The claim is evidenced by testing the confidence/accuracy over encoder-only language models like mBERT and XLM-R, and decoder only  models like Llama2.

- The claim is evidenced by testing over in-distribution and Out-of-Distribution (OOD) Tasks

For the second claim:

-The claim is evidenced by testing the data augmentation strategy, basically translate from english to other languages.

- The authors also test other techniques, e.g., in-context learning, Temperature Scaling.

**Requested Changes:**

I am not a expert of model calibration, and I am not a fan of this research topic as well. I find that this research topic does not make much sense to me honestly, and I can't see its research value. Please try to convince me by providing evidence to answer my following question:


1. (major) Is making the model calibrate between confidence and accuracy really a different problem than only improving the accuracy of the model on specific task? Let say the model has high accuracy on one task, is that possible that it is ill-calibrated, i.e., its accuracy does not match its confidence? I want to raise the question because from your conclusion, if we finetune on Chinese data, then its calibration will improve. Isn't this improve of calibration is just because the model's accuracy on Chinese (which you finetune on) is improved? If that's the case, it seems that the reason leading to better calibration of these three strategies, i.e., Data Augmentation, in context learning. etc is obvious, because they all improve the accuracy of the corresponding language tasks (e.g., Chinese).

2. What is the point of calibrating a model? How does it generally improve the service performance of the model?

The paper is just fine with a lot of evaluation results. But I could not see any novelty and something that interest me in general. However, based on TMLR criterion, we should not reject a paper based on novelty but should focused on empirical evidence. Therefore, I will not stop your publication if you can answer my sort of "mean" questions.

---

> ### Author Response · Authors · 2025-09-06
> **Response to Reviewer VPdc**
>
> Thank you for your feedback. We address your questions below, focusing on the distinction between accuracy and calibration and the practical value of our work.
>
> **Accuracy vs. Calibration: A Technical Distinction**:\
> Yes, calibration is a fundamentally different problem from accuracy, both conceptually and empirically.
> Conceptually, the two metrics answer different questions. Accuracy asks, “How often is the model’s top-ranked prediction correct?” In contrast, calibration asks, “When the model reports X% confidence, is it correct X% of the time?”. A model can achieve high accuracy while being poorly calibrated—for instance, by being systematically overconfident. This misalignment makes its probability scores unreliable for practical tasks like setting abstention thresholds.
> Empirically, our work demonstrates several instances where accuracy and calibration decouple:
> - Temperature Scaling (TS) improves ECE at a fixed EM. TS is designed to adjust confidence scores without altering the final predictions, thereby leaving accuracy unchanged. Our results show this clearly: for mBERT on XQuAD, applying TS on a merged dataset drops its ECE from 16.36 to 5.86 while EM remains fixed at 46.77. Similarly, for mT5, ECE falls from 23.06 to 10.19 with EM constant at 50.95.
> - Data augmentation improves calibration far more than accuracy. For mBERT, augmenting the fine-tuning data with translations (En-tr) increases EM modestly from 43.51 to 46.69 but slashes ECE from 17.75 to 5.58. This shows a calibration improvement that is disproportionately larger than the gain in accuracy.
> Calibration degrades faster than accuracy across languages. In the zero-shot cross-lingual setting, we found that the average prediction error for non-English languages increased by about 70% relative to English. However, the average ECE increased by a much larger 145% (from 7.32 for English to an average of 18 for other languages), highlighting a distinct and more severe calibration gap.
> - ICL shows a distinct calibration benefit. For LLaMa2 on TyDiQA, using an adaptive 2-shot ICL strategy improves both EM and ECE. Crucially, it decreases ECE by more than 10% compared to a random ICL strategy, demonstrating a meaningful improvement in calibration that is not solely a byproduct of improved accuracy.
>
> **The Practical Value of a Calibrated Model**:
> - Calibrated confidence scores unlock critical operational capabilities that accuracy alone cannot provide.
> Reliable Abstention and Triage. In many services, a confidence threshold is used to decide when a model should abstain from answering and escalate to a human. If a model's confidence is not calibrated, the same threshold will correspond to different levels of real-world accuracy for different languages, leading to inconsistent performance. Our work shows that applying TS on a small, mixed-language validation set effectively calibrates the model, making these thresholds more reliable across languages.
> - Cross-Lingual Fairness and Trust. We show that calibration error is highly correlated with linguistic distance from English and the amount of pre-training data. This means models are often most overconfident—and least reliable—for the very languages and user groups that are already underserved. Improving calibration narrows this gap, ensuring that a reported "80% confidence" has a consistent meaning, whether the user is in the US or in a low-resource language community.
> - Actionable User-Facing Reliability. Reliability diagrams, like the one we show for Bengali in Figure 1, provide a clear visual diagnostic of a model's trustworthiness in a specific language. This evidence allows teams to implement language-aware guardrails and manage user expectations effectively.
>
> **Direct Answers to Key Questions**:
> - "If we fine-tune on Chinese and calibration improves, isn’t that because EM improved?"\
> Not necessarily. We provide a direct counterexample:
> Temperature Scaling improves ECE substantially while EM is held constant. This demonstrates a pure calibration effect, independent of accuracy gains.
> - "Is calibration really a different problem?"\
> Yes, both by definition and empirically. As shown above, ECE degrades and improves differently than EM under various conditions (cross-lingual transfer, data augmentation, etc.). We also report on additional calibration metrics and reliability diagrams, reinforcing that we are measuring a distinct model property.
> - "What’s the point in practice?"\
> Calibrated models enable coverage-controlled QA (consistent abstention across languages), safer deployment (fewer overconfident errors in low-resource languages), and robust threshold transfer from development languages to unseen ones, as shown by our mixed-language TS results.

---

> ### Comment · Reviewer_VPdc · 2025-09-11
> **Response**
>
> Hi authors,
>
> I could not understand why temperature scaling can lead to better calibration. Can you use a more intuitive way to convince me why increase/decrease temperature of output lead to better match of accuracy and confidence? To me, it seems that uniformly increase tempertures make the output probability more smooth, I.e., prediction logit (I.e., confidence) decrease for both accurate answer and non-accurate  answer but I can't see why it leads to better calibration (I.e., lower ECE score see your Section 3.2 ).

---

> > ### Author Response · Authors · 2025-09-22
> >
> > Your intuition is correct: temperature scaling uniformly adjusts the output distribution. It improves calibration because modern networks are systematically overconfident, and the temperature T is a learned parameter that corrects this specific bias.
> >
> > Here's the technical breakdown of why this reduces the Expected Calibration Error (ECE):
> >
> > - The Problem: Overconfidence from Loss Minimization. During training, minimizing cross-entropy loss incentivizes the model to make the logit for the correct class infinitely larger than others. This pushes the softmax output towards 1.0, creating a systematic gap where a model's average confidence (e.g., 95%) is higher than its empirical accuracy (e.g., 80%). This disparity is the primary source of high ECE.
> >
> > - The Mechanism: Rescaling the Logit Space. Temperature scaling with $T > 1$ divides all pre-softmax logits by $T$. This operation contracts the logit space, reducing the relative distances between them. The resulting post-softmax distribution is "softer" (higher entropy), with probabilities pulled away from the extremes of 0 and 1. This directly counteracts the overconfidence bias.
> >
> > - The Optimization: Principled Correction via Validation. The temperature $T$ is not an arbitrary hyperparameter; it is optimized to minimize the Negative Log-Likelihood (NLL) on a held-out validation set . This optimization process finds the specific scaling factor $T$ that best realigns the model's posterior probabilities with the empirical correctness rates observed in the validation data. In essence, the optimization finds the precise $T$ to correct the model's systemic confidence error. By applying this learned $T$, overconfident predictions are scaled down into confidence bins that more accurately reflect their true probability of being correct, thus minimizing the |accuracy-confidence| term in the ECE calculation.

---

### Review · Reviewer_hivZ · 2025-08-23

**Summary Of Contributions:**

The paper presents an empirical study of confidence calibration in multilingual QA models across extractive (mBERT, XLM-R), encoder–decoder (mT5, mBART), and decoder-only (LLaMA-2-7B) architectures. Models are fine-tuned on English (SQuAD 1.1) and evaluated zero-shot on XQuAD/MLQA as well as under distribution shift on TyDiQA. The authors measure calibration with Expected Calibration Error (ECE) and exact match (EM). They then test the effect of interventions on confidence calibration: temperature scaling (TS), label smoothing (LS), few-shot cross-lingual fine-tuning via machine-translated data, and in-context learning (ICL) for LLMs. The authors find that (a) calibration on English does not transfer well to other languages; (b) TS tuned on a small multilingual validation set consistently outperforms tuning on a larger English-only set; (c) adding machine-translated samples during fine-tuning improves cross-lingual calibration; and (d) ICL improves both EM and ECE for LLaMA-2 on TyDiQA.

**Strengths:**
- The paper provides a large empirical study across multiple model types, languages (both high and low-resource), and datasets.
- The paper provides practical and actionable findings, such as temperature scaling on multilingual validation data being more effective than English-only validation, and the benefits of incorporating translated data during fine-tuning.

**Weaknesses:**
- The paper studies only a single decoder-only LLMs. It would be informative to replicate ICL results at larger scales and compare to other multilingual-capable LLMs, such as Aya-23.
- All models are fine-tuned exclusively on English SQuAD before evaluation. While this setup cleanly isolates cross-lingual transfer, it may also introduce artefacts: improvements from translation-based augmentation could partly reflect recovery from an artificial bottleneck introduced during fine-tuning.

**Audience:**

Yes

**Audience Explanation:**

The findings are relevant for researchers and practitioners working on multilingual NLP, calibration, and QA systems. The paper offers practical insights, such as the effectiveness of multilingual validation for temperature scaling, benefits of small-scale translated data, and ICL improvements.

**Claims And Evidence:**

Yes

**Claims Explanation:**

The main claims are supported by sufficient empirical evidence. For example, the finding that calibration on English does not transfer to other languages is demonstrated in Table 1. The benefits of temperature scaling tuned on small multilingual validation sets over larger English-only ones is shown in Table 3. The benefits of adding translated samples is demonstrated in Table 4 and Table 5. ICL results for LLaMA-2 on TyDiQA show improved EM and ECE in Figure 4, though they are limited to a single model.

**Requested Changes:**

**Critical for recommendation:**
- Extend the ICL experiments beyond LLaMA-2-7B to other multilingual models such as Aya-23.

**Strengthen the work:**
- Analyze how calibration results depend on candidate set size K and decoding strategy (greedy, beam, sampling), especially for generative models.
- Add statistical significance tests to support claims of improvement, particularly where ECE differences are small.

---

> ### Author Response · Authors · 2025-09-06
>
> Thank you for your detailed review and suggestions. In direct response, we have extended our experiments to the latest Aya model, Aya-expanse-8B. Following the same setup used for LLaMA-2-7B, we fine-tuned Aya-expanse-8B on the SQuAD dataset and evaluated its calibration and accuracy on XQuAD and TyDiQA. For TyDiQA, we ran in-context learning experiments with both RANDOM and Adaptive demonstration selection, as we did for LLaMA-2-7B in Section 5.3.
>
> On XQuAD, the calibration performance for non-English data is comparable to or better than the performance on English. Across both datasets, we therefore observe that Aya-expanse-8B provides improved non-English calibration compared to other models, which supports that when a model is optimized for multilingual data such as Aya, non-English calibration also improves.
> - The updated Figure 3 and Table 5 in the manuscript will include the Aya results.
>
> | | **en** | | **ar** | | **de** | | **el** | | **es** | | **hi** |\
> | **XQuAD** | EM | ECE | EM | ECE | EM | ECE | EM | ECE | EM | ECE | EM | ECE |\
> | **FT** | 84.71 | 10.13 | 68.66 | 7.93 | 70.25 | 5.21 | 66.64 | 7.37 | 76.89 | 8.06 | 66.13 | 4.33 |\
> | | **ro** | | **ru** | | **th** | | **tr** | | **vi** | | **zh** |\
> | **XQuAD** | EM | ECE | EM | ECE | EM | ECE | EM | ECE | EM | ECE | EM | ECE |\
> | **FT** | 75.54 | 5.78 | 62.77 | 9.84 | 54.71 | 5.35 | 64.12 | 10.96 | 71.85 | 8.13 | 73.28 | 4.67 |
>
>
> For TyDiQA, we additionally experiment with the RANDOM and Adaptive in-context learning settings as in Section 5.3 and show that in-context learning improves confidence calibration. This observation is consistent with our earlier results on LLaMA-2-7B, where in-context learning provided an efficient way to improve calibration for decoder-only language models. However, we find that in-context learning doesn’t always improve accuracy in this case. Note that we report results with the base Aya-expanse-8B (not a fine-tuned variant) here as in-context learning slightly reduced accuracy in our setting.
>
> | **TyDiQ A** | **English** | | **Arabic** | | **Finnish** | | **Indonesian** | | **Korean** | | **Russian** | | **Swahili** |\
> | | **EM** | **ECE** | **EM** | **ECE** | **EM** | **ECE** | **EM** | **ECE** | **EM** | **ECE** | **EM** | **ECE** | **EM** | **ECE** |\
> | **English** | 70 | 21.21 | 70.58 | 13.57 | 57.8 | 17.49 | 74.34 | 15.62 | 82.61 | 18.97 | 63.3 | 16.21 | 42.08 | 18.79 |\
> | **Random** | 69.32 | 6.95 | 72.53 | 16.42 | 56.91 | 14.19 | 75.22 | 8.86 | 81.16 | 16.2 | 62.93 | 14.44 | 66.93 | 10.27 |\
> | **Adaptive**| 70.68 | 11.01 | 71.34 | 12.93 | 67.77 | 9.44 | 77.17 | 9.08 | 81.88 | 13.59 | 66.75 | 12.66 | 66.93 | 15.67 |
>
> We will further evaluate the impact generation configurations to confidence calibration and add statistical tests in our experiments for the final version of the paper.

---

### Author Response · Authors · 2025-09-06
**Response to Reviews**

We thank the reviewers for their constructive feedback and for appreciating the impact of our paper.
We have provided detailed responses with added experiments for the Aya model. Please let us know if there are any other questions.
We believe our comprehensive empirical study on multilingual calibration for QA will be greatly valuable to the research community.

---

### Decision · Action_Editor_937R · 2025-11-19

**Recommendation:** Accept with minor revision

**Additional Comments:**

Please add more thorough and intuitive explanation on  why different strategies, e.g., incorporating a small set of cheaply translated multilingual samples during fine-tuning/calibration effectively enhances the calibration performance, or why temperature scaling lead to better calibration.

**Audience:**

Yes

**Audience Explanation:**

Reviewers acknowledged that the paper's empirical findings could be of interest for researchers and practitioners working on multilingual NLP

**Claims And Evidence:**

Yes

**Claims Explanation:**

Reviewers generally agreed that the paper presentation and results are sound, and the claims made in the paper are supported by the results.